# Can Shared Control Improve Overtaking Performance? Combining Human and Automation Strengths for a Safer Maneuver

**DOI:** 10.3390/s22239093

**Published:** 2022-11-23

**Authors:** Mauricio Marcano, Fabio Tango, Joseba Sarabia, Silvia Chiesa, Joshué Pérez, Sergio Díaz

**Affiliations:** 1TECNALIA, Basque Research and Technology Alliance, 48160 Derio, Spain; 2Centro Ricerche Fiat, 10043 Orbassano, Italy; 3Faculty of Engineering, University of the Basque Country UPV/EHU, 48013 Bilbao, Spain; 4RE:Lab, 42122 Reggio Emilia, Italy

**Keywords:** shared control, arbitration, human–machine cooperation, human–machine interface, overtaking, safety, automated driving

## Abstract

The Shared Control (SC) cooperation scheme, where the driver and automated driving system control the vehicle together, has been gaining attention through the years as a promising option to improve road safety. As a result, advanced interaction methods can be investigated to enhance user experience, acceptance, and trust. Under this perspective, not only the development of algorithms and system applications are needed, but it is also essential to evaluate the system with real drivers, assess its impact on road safety, and understand how drivers accept and are willing to use this technology. In this sense, the contribution of this work is to conduct an experimental study to evaluate if a previously developed shared control system can improve overtaking performance on roads with oncoming traffic. The evaluation is performed in a Driver-in-the-Loop (DiL) simulator with 13 real drivers. The system based on SC is compared against a vehicle with conventional SAE-L2 functionalities. The evaluation includes both objective and subjective assessments. Results show that SC proved to be the best solution for assisting the driver during overtaking in terms of safety and acceptance. The SC’s longer and smoother control transitions provide benefits to cooperative driving. The System Usability Scale (SUS) and the System Acceptance Scale (SAS) questionnaire show that the SC system was perceived as better in terms of usability, usefulness, and satisfaction.

## 1. Introduction

Intelligent agents have been entering our lives and supporting us in a wider variety of tasks, attracting growing attention. This has been especially true for automated driving (AD) technologies due to their significant advantages. They include better convenience and greatly reduced congestion costs, but particularly an improvement in road safety by mitigating the consequences of human errors while driving (e.g., drowsiness, distraction, reckless driving, or speeding).

### 1.1. The Road Safety Problem

The traffic safety reports from the National Highway Traffic Safety Administration (NHTSA) have registered more than 35,000 injury crashes, where around 8% correspond to crashes due to driver misbehavior. As a result, over 3100 people died, and another 320,000 people were injured [1].

Concerning crashes, head-on collisions are one critical type of accident as the likelihood of serious injuries and fatalities is high. A common scenario where such collisions occur is overtaking on two-way roads (which represent about 90% of the Spanish road network [2]). In this sense, though improper overtaking is not the major cause of accidents, overall, it does represent a high percentage of fatalities when accidents occur (in Spain, accidents involving heavy trucks overtaking have the highest percentage of fatalities, which is around 20% [3]). In addition, in Italy, of the total of overtaking accidents, more than 30% present injury outcomes. Moreover, according to the Spanish General Direction of Traffic (DGT), 40% of accidents on two-way roads are due to off-lane events, and 27% are because of head-on collisions (Spanish General Agency of Traffic (DGT): https://revista.dgt.es/es/noticias/nacional/2019/01ENERO/0128cambio-normativo-velocidad-90.shtml, accessed on 26 July 2022). Another recent statistic (2020) from the NHTSA [4] shows that of the total of accidents in the US that result in driver fatalities, 60% comes from head-on collisions.

### 1.2. The AD Solution

From this perspective, AD systems are becoming a technological solution to improve road safety and reduce fatalities. In comparison with human drivers, automation is capable of faster response times, can handle greater amounts of information, and process it more quickly; therefore, AD promises to improve safety by removing human drivers from the control loop (see the SAE taxonomy for the levels of automation [5]).

However, despite great advantages, Automated Vehicles (AVs) still have situations where such systems reach their limits and will not be able to work reliably. For example, automation cannot cope with highly complex traffic situations (e.g., dense urban traffic, for perception difficulties) and can have the challenges of legal restrictions and accident responsibility.

Although AVs aim at revolutionizing our “consolidated concept” of transportation, at the same time, they introduce new challenges by removing the human from the control loop, leaving to the driver the monitoring/supervision task. However, under this role, there is a reduction in human workload that presents two potential drawbacks. On the one hand, a high probability of error (when resuming control of the vehicle). On the other hand, there is also the risk that humans lose some driving skills. Therefore, intelligent and advanced interaction methods are crucial to enhance user experience, acceptance, and trust, which means eventually safety.

With conditional AVs (SAE-L3), drivers are no longer required to actively monitor the driving environment and can be allowed to fully engage in non-driving-related tasks (NDRTs). Some studies have shown that this limited driver–vehicle interaction could produce in drivers a loss of situational awareness [6] or even lead them to become drowsy [7], increasing the difficulty to take over control safely when requested. For partial AVs (SAE-L2), the problem is that the driver over-trusts (or misuses) automation and stops monitoring the environment and being ready to take control at any time (accidents have been reported in L2 AVs [8]).

### 1.3. The Shared Control Solution

To avoid the drawbacks of automation, minimize its disadvantages, and simultaneously maximize its advantages to support the driver’s decision, we propose in this paper the Shared Control (SC) cooperative scheme, where driver and automation drive together and simultaneously at the tactical and control level [9,10], thus taking the advantages of both the human driver and the automated system.

Under this mode of cooperation, we should consider the capabilities and limitations of the individual parties, but also the quality of their cooperation (such as described by the joint cognitive system of Woods & Hollnagel [11]). In this sense, this work presents an evaluation of a solution based on shared control for the scenario of overtaking on two-way roads, in which driver and automation cooperate in all the hierarchical driving levels as a team [12], especially in decision and control.

### 1.4. Current Research

Different Advanced Driver Assistance Systems (ADAS) have been proposed in the literature to improve safety while overtaking on two-way roads. One example is the See-Through-System [13], a solution based on augmented perception. Here, the driver can see through the front vehicle, which is equipped with a vision camera in the front and screens in the rear showing the road ahead to the vehicle behind (this solution has been implemented in Samsung trucks). There are also cooperative systems where the driver communicates with the automation through tactical commands, such as the system presented by Walch [14] used in the same overtaking scenario. In this work, the collaboration is undertaken through a Human–Machine Interface (HMI) to approve or reject the overtaking maneuver, but the automation performs all the lateral and longitudinal control. As for shared control solutions that involve the driver at the operational level, few works address similar scenarios. Ercan [15] and Muslim [16] present collision avoidance systems for dangerous lane changes due to the blind spot (i.e., the scenario considers vehicles coming in the same direction). Nishimura [17] evaluates the situation of overtaking, but with a focus only on the control transitions from automated to manual and vice versa, but not on steering correction. In another work, Dillman [18] presents a comparison of overtaking modes, including a semi-automated one based on shared control, where the automation performs an overtake as soon as the driver puts the hands on the steering wheel and confirms the intention to overtake. However, to the knowledge of the authors, the current literature does not present a system based on shared control that handles control transitions and avoids dangerous overtaking maneuvers on roads with oncoming traffic.

### 1.5. Our Contributions

Our work explores the advantages of the shared control adaptive co-pilot over commercially available autopilots (i.e., SAE L2 systems combining Automated Lance Centering (ALC) with Adaptive Cruise Control (ACC) functions), by evaluating both systems with real drivers in a Driver-in-the-Loop (DiL) simulator. The SC solution includes three main innovations:A steering control strategy that provides smooth and context-dependent transitions for the de-activation and the re-activation of the automation.Steering torque corrections to avoid collision with oncoming vehicles (with an intensity of correction proportional to the driving risk indicator).An HMI that informs the driver about the system limitations and the suggestion to take control and overtake.

The rest of the paper is structured as follows. Section 2 presents the methodology applied to the experimental study. Section 3 provides details regarding the technical developments of the AD algorithms (especially the arbitration system and shared controller), together with the HMI design. Section 4 provides detailed results of the experiment, both objectively and subjectively, finishing with a discussion of the results in Section 5 and closing with conclusions and lessons learnt in Section 6.

## 2. Methodology

This section presents the methodology used for the studies conducted in the Driver in the Loop (DiL) simulator with human drivers participating directly in the experimental phase. The study focuses on a scenario where the adaptation between human and machine agents is fundamental for safe and effective operation in AD. The methodology is ordered in terms of the use case (the real scenario where the system functions are useful), the experimental setting and apparatus (simulator and its components), and the experiment design (with details on participant profiles, experimental conditions, and the procedure to conduct the tests).

### 2.1. Use Case

As part of the developments of the PRYSTINE project [19], different use cases (UCs) were elaborated and implemented to investigate the driver–automation team concept. In particular, the present study is aligned with the UC named “driver–automation cooperation in overtaking” (see Figure 1). It illustrates a transition from automated to manual mode, initiated by the driver, to support the system during an overtaking maneuver (which, however, also supports the driver by giving steering corrections to preserve safety). Therefore, mutual support is present in the UC. The related narrative is as follows:


*“On an extra-urban road, Silvano is driving an L2 vehicle in AD mode to come back home (at 90 km/h), when the vehicle approaches a big and slow truck ahead, driving at 70 km/h. The ADS cannot overtake due to the limited perception (it cannot “see” and check if other vehicles are coming in the opposite direction and thus if the left-lateral lane is free for the maneuver). In such a situation, the ADS can only slowly follow the vehicle in front, waiting until it changes its route. However, Silvano is in a hurry due to an appointment for dinner and thus he is getting nervous. In this sense, the system based on shared control asks Silvano for support in case he wants to overtake”.*


### 2.2. Experimental Settings and Apparatus

The functionalities developed for the scenario were tested on a DiL simulator (see Figure 2), which consists of a cockpit, the driving control mechanisms, the displays, and the control PC. The driving actuators are the key components of the simulator. On one side, the steering wheel consists of a model 130ST servo motor with a maximum rated torque of 15 Nm. It is equipped with an incremental encoder and a current sensor. On the right side of the steering wheel, there is an emergency button for safety purposes. There is also a brake pedal and an accelerator pedal with different pressure and damping coefficients, simulating the feel of some pedals in real vehicles. The driving environment is displayed on three front screens (26 inches each). Apart from them, a 15.6” screen has been placed behind the steering wheel to display a full digital HMI connected to the computer to display generated graphic components for the communication of the overtaking request. The vehicle simulation software is Dynacar, working in combination with MATLAB/Simulink for the development of the AD framework [20].

### 2.3. Experiment Design

A within-subject experiment took place with a total of 13 participants, testing two experimental conditions, and driving around 10 min each of the two systems.

#### 2.3.1. Participants

A total of 13 participants took part in the study (6 Females, 7 Males), aged 23–64 years (mean = 35.6, SD = 13.0), all of them with at least 2 years holding a driving license. The experiment was conducted with the consent of each participant.

#### 2.3.2. Experimental Conditions

The challenge in the scenario is represented by the limits in the ODD of the system, for which it is impossible to see beyond the obstacle (a big truck) due to its limits in perception, so no overtaking maneuver is performed by automation. In this context, the baseline is a vehicle with AD functionalities (L2). As an alternative, a vehicle with the same automation level as the baseline, but with the addition of shared control (SC) functionalities is proposed. In this case, if the system “knows” that the driver prefers overtaking (e.g., “being in a hurry”), it proposes cooperation in perception and/or inaction [21] to perform the overtaking maneuver, giving continuous support to the driver. The differences between L2 and SC conditions are illustrated in Figure 3 and detailed below:L2: As a baseline, an ADS with ALC and ACC activated is tested. When the driver decides to overtake, the system deactivates when the steering torque exceeds a 3 Nm threshold. A sound confirming deactivation is heard. During the takeover, the driver is free to drive, without any system intervention. Once the overtaking is finished and the conditions allow it, the driver needs to re-activate the system by pressing the button on the steering wheel and a confirmation sound is heard. A beep sound is issued when the ego-vehicle has spent a few seconds behind the truck as an indication to overtake, which is used only to have consistent overtaking attempts in the two experimental conditions.SC: As part of the proposed system, shared control functionalities are added over the baseline condition. The system is deactivated when there is enough field of view and if there is enough time to do the overtaking (not only by torque effort). In addition, the system reengages the automated mode without any explicit indication from the driver when returning to the right lane (not manually). If there is not enough time to overtake, the system increases the steering correction torque in proportion to the risk of collision to abort the maneuver. An HMI indicates to the driver that the system needs support to overtake and informs about the automation state.

#### 2.3.3. Procedure

Participants filled out a pre-drive questionnaire related to demographic data and driving profiles. In addition, they familiarized themselves with the simulator, taking a driving session in manual mode. The study was a within-subject experiment, and the order of execution of both systems was different according to the two corresponding participant groups. Figure 4 shows the schematic of the experiment procedure including the order of application of questionnaires.

In the official test, participants drove on a straight road. The scenario involved sequential overtaking maneuvers. The experiment was designed so that each driver faced oncoming vehicles when attempting to pass the truck in four situations (as shown in Figure 5): (1) no oncoming vehicle appears, (2) the vehicle becomes visible at a far distance, with enough time to overtake; (3) the vehicle becomes visible at a mid-range distance with no time to overtake; (4) the vehicle becomes visible at a very short distance, requiring a fast aborting maneuver.

The speed of the truck is 70 km/h, the same as the speed of the oncoming vehicle, while the speed set for the ACC of the ego-vehicle is 90 km/h (the maximum speed on the road). In addition, in manual mode, the system does not limit the vehicle speed above 90 km/h so the driver can exceed the limit if necessary for safety reasons. The driver was instructed that the goal was to maximize the distance traveled (i.e., overtake as fast as possible, but without accidents) for the duration of the tests.

## 3. System Design

The shared control system (shown in Figure 6) is composed of two main sub-modules: the arbitration module, and the shared controller [22]. The arbitration system decides in real time how to distribute the control authority between the driver and the automation. The shared controller corresponds to the operational level, which executes the control actions, applying the calculated authority to the automated system mechanism (steering wheel). Some authors also refer to a third level, commonly named the execution level [9], which is a torque controller that is integrated into the steering wheel device that supplies the steering motor with the desired current. In this sense, the hardware of the automated steering system has specific requirements to support the driver appropriately (provide more than 5 Nm peaks, a control frequency between 1 and 10 ms., and variable damping configuration to include the stability criterion from Equation (7)). The specific algorithms used in the tactical and operational levels of the shared-control framework are explained below, focusing on the specific use case of driver–automation cooperation in overtaking.

### 3.1. Arbitration

A co-driving system can effectively improve vehicle safety and reliability through the shared control scheme. It is composed of the following elements: the human driver, automation, vehicle, and environment. The driver is one of the key links in this “team”, with the related driving characteristics and states. Given this context, traditional system modeling–based on conventional mathematical tools–is not well-suited for dealing with ill-defined problems and uncertain systems. By contrast, a Fuzzy Inference System (FIS) employing fuzzy if-then rules can model the qualitative aspects of human expertise. The main reason to use a FIS is the need for a model based on the human expert knowledge of the current situation to effectively assess the risk.

Fuzzy Logic (FL) approximates human reasoning and does a good job of balancing the tradeoff between precision and significance (as illustrated in Figure 7). It is a convenient way to map an input space to an output space. In addition to that, we took this direction in our work to offer an approach closer to the so-called Explainable-AI (XAI), than the “more traditional” AI–such as Deep Learning (DL), which is a kind of black box. The basic concept of FL is to deal with linguistic variable(s)–that is, a variable whose values are words rather than numbers. Although words and labels are inherently less precise than numbers, their use is nevertheless closer to human intuition. Moreover, computing with words exploits the tolerance for imprecision and uncertainty. In this sense, our approach facilitates the interaction process with the driver, where human experiences are relevant: FL controllers have been chosen for their intuitive tuning and easy implementation, which can reduce the gap that exists, so far, between AVs and human drivers.

To sum up, FIS may be viewed as the principal constituent of so-called soft computing (towards the direction of XAI), whose goal is exactly to accommodate the imprecision of the real world. From this point of view, soft computing could play an increasingly important role in the conception and design of automated systems, especially if the interaction with the human factor is required and considered. In detail, the arbitration module emulates a human driver’s behavior, with information on external variables related to the driver, vehicle, and environment.

For our shared control system, applied to the overtaking maneuver, the arbitration module is responsible for negotiating the appropriate haptic authority of the lateral controller. It decides, depending on the different driving variables, whether the controller should be given high authority (the automation has the main control) or low authority (the driver applies most of the required torque). In this sense, this module is also responsible for control transitions, in both directions (automated to manual and vice versa).

For the design of the arbitration module, three fuzzy inputs have been included, i.e., vehicle position, driver’s intention, and maneuver risk, such that this decision-making system makes a smooth transition of the level of authority. The fuzzy technique considers three stages: fuzzification, inference, and defuzzification. The input values are transformed and interpreted as fuzzy data in the fuzzification stage processes. Each variable is defined by a membership function involving its corresponding linguistic labels, represented in the inference process, due to the rule base defined (see Table 1). Finally, the defuzzification operation converts a fuzzy value into a classical value specified in the output membership function (level of authority). Each variable is illustrated in Figure 8, and defined as follows:**Vehicle position***:* Represented as the lateral error of the vehicle to the center of the right lane. The labels of the membership functions (*Right, Border,* and *Left*) represent the different positions of the vehicle on a two-lane road.**Driver intention**: Represented as the derivative of the lateral error of the vehicle. The labels of the membership functions (*Away, Stay,* and *Return*) represent the driver’s intention to leave the lane, stay in the same direction, or return to the lane. This intention is combined with the lateral error to obtain an estimate of the lane change intention.**Maneuver risk:** Represented as the distance-to-collision between the vehicle and the oncoming vehicle in the left lane. The labels of the membership functions (*Far* and *Close*) represent the relative distance between the two vehicles, indicating low and high collision risk, respectively.**Level of Authority:** Represented as the maximum steering torque of the correction in Nm. The labels of the membership functions (*Manual, Assistance,* and *Override*) represent the full range of automation steering assistance from none to gentle corrections, to maximum assistance that can even exceed the force exerted by the driver.

The fuzzy output is then proportionally combined with the normalized field of view of the ego vehicle, to consider the automation perception capability within the arbitration.

### 3.2. Shared Controller (Operational Level)

According to a recent state of the art on control techniques for shared-control applications in automated driving [10], lateral controllers should meet at least three requirements: (1) to control the steering mechanism by torque, to improve the interaction with the driver’s hands actions, (2) to use an optimal control method, to balance the multiple objectives of shared-control applications (e.g., comfort, safety, performance, and efforts), and (3) to have ana adaptive haptic authority, to assist the driver with different torque intensities according to the demands of the driving context.

In this sense, the controller implemented for this experimental study is a torque-based lateral vehicle controller based on Non-Linear Model Predictive Control (NMPC), which has been previously published in [12], for the application of support to a distracted driver. The main components of the controller are described below.

**Minimization of objectives:** Three main objectives are defined (as in Equation (1)). First, the tracking performance ($J_{t})$, associated with vehicle position to the reference trajectory. Second, the driving comfort ($J_{c})$, linked to the steering wheel angular velocity (w) and the vehicle yaw rate (ψ). Third and last, the steering effort, related to the torque control signal ($T_{mpc})$ and its rate change ($\Delta T_{mpc}$), is represented as the input optimization function $J_{u}$.


(1)
$$J=J_{t}+J_{c}+J_{u}$$


**Constraints of variables:** The ability of NMPC to add states and inputs constraints allows the limiting of values that can help in the preservation of safety, for example, a constraint in the yaw rate is included to avoid the vehicle drifting after the correction maneuver (Equation (2)). Additionally, constraints are added to the control variable for effort and comfort purpose (Equations (3) and (4)).


(2)
$$\left|\psi_{max}\right|\;\leq 0.5\;\mathrm{rad}/s$$



(3)
$$\left|T_{max}\right|\;\leq \;\lambda$$



(4)
$$\left|\Delta T_{max}\right|\leq 2\;\mathrm{Nm}/s$$


**Adaptive authority:** To assist the driver with variable intensities of torque, the level of haptic authority is included in the road-vehicle model through the torque derivative equation (Equation (5)). In addition, the authority is linked to the maximum steering torque provided by the control through the constraint (Equation (3)).


(5)
$$\dot{T_{mpc}}=\lambda \Delta T_{mpc}$$


**Stability Criterion:** Adding authority to the controller is equivalent to increasing its stiffness, therefore the increase in the authority makes the system prone to oscillations. In this sense, a stability criterion is designed to keep the NMPC stability for different values of λ. It is related to the value of an equivalent damping value of the steering motor as in Equation (6).


(6)
$$b_{e}=\sqrt{\left(\lambda +1\right)/2}$$


For the baseline condition (i.e., automated vehicle L2), the ALC controller is the torque-based NMPC configured with λ=3 Nm, while the ACC function uses a variable speed reference, fuzzy logic controller. No arbitration module is used because the ALC controller always works with a fixed authority. For the transitions of control, the deactivation occurs when the driver exerts more than 3 Nm of torque, while the activation is performed by pressing a button on the steering wheel. Both transitions use a ramp with a duration of 0.3 s.

Unlike the baseline (fixed authority), the shared-control mode applies a variable authority calculated by the arbitration module. Due to the aggressive nature of the maneuver, the NMPC design also includes the yaw rate constraint to prevent excessive vehicle drift that could lead to instability and accidents (Equation (2)).

### 3.3. Human–Machine Interface

Different scientific findings on partial AVs highlight the importance of the HMI designs to provide feedback about the behavior of the vehicle [23]. They explain the actions that the vehicle is undertaking and the monitoring of the status of the vehicle, by providing information about the vehicle’s performance. A systematic review of HMI design principles that promote trust in AD systems [24] underlines the fundamental importance to inform users about the situations and events that are about to happen.

The current human-centered automation design focused on creating an environment in which humans and machines can work together cooperatively has aroused increasing interest. The principles underlying this design approach highlight, among others, criteria such as the usability of the technological interfaces, the human control of automated activities, the dissemination of appropriate information, the participation of workers and customers in continuous improvement, and much more. The design of the HMI is important in the condition of shared driving, where the systems have to communicate to the driver the human–machine cooperative status [25] as well as in the control-transition feature in semi-autonomous vehicles.

For all levels of driving automation, it is important to keep a certain ‘transparency’ of the automated system so that the users can keep the sense of being “in charge” [26]. To communicate the overtaking request and the vehicle state to the driver, a visual HMI has been developed. Based on the results of the literature analysis discussed previously, the developed HMI continuously provided feedback that explains the vehicle’s choices and an effective communication style that presents the system has been adopted.

The HMI displays information about all the relevant elements in the environment, including the lateral lane and the truck in front of the vehicle. The shared driving mode is the main vehicle state mode in which the automated system oversees the longitudinal control, while the lateral control is shared between the human driver and the machine. In this condition, the system can dynamically retrieve and convey the authority in accord with the overall conditions of the vehicle to prevent and avoid any possible road accident. The color of the HMI elements changes to distinguish the state of the vehicle.

In the HMI, the speedometer and the RPM with the current gear indicator do not change to the manual mode since in the shared condition the driver is still involved in the driving task. The dials on the sides, the speedometer, and the tachometer are designed to retrieve all the most important information in a small space and are always given to the driver.

The motivation for any change performed by the arbitration control system is conveyed by the system. Moreover, information regarding the level of shared lateral control is provided through a bar. The bar represents the level of automation in a percentage: the higher the dimension of the bar, the higher the level of lateral control taken by the vehicle.

Since the scenario contains a case of a non-stable state which requires cooperation in the decision to overtake a truck in a situation of limited sensors’ visibility, the HMI indicates to the driver that the system needs support to overtake and informs them about the automation state through a pop-up message and a warning signal.

The action required of the human driver is to take the latter control through the steering wheel but is not mandatory, because without any action the vehicle would simply continue to follow the track ahead. If the required action is performed, the vehicle will switch to shared driving mode. Figure 9 shows the sequence of the maneuver and its corresponding representation in the visual HMI. The lanes highlighted in green indicate that the automated mode is active, and no color represents the manual mode. The bar indicates how the control authority is distributed.

## 4. Data Analysis and Results

This scenario has a strong safety component since the main objective is to avoid accidents and unsafe events during the overtaking maneuver. Since the developed systems consist of a strategy to operate in different modes during driving (automated, manual, transition), the first category of Key Performance Indicators (KPIs) evaluates the proportion of time that the system operates in each driving mode. Then, the second category of quantitative KPIs focuses on the safety evaluation of the system.

The experimental data were collected from two sources. First, the objective data, related to the vehicle and the driver, are collected at a frequency of 1 ms. through the simulation environment, composed of the vehicle dynamics software Dynacar and the development environment Matlab/Simulink. Second, the subjective data, related to the driver’s perception of the system, are collected through standard questionnaires, particularly the System Usability Scale (SUS) [27,28], and the System Acceptance Scale (SAS) [29].

### 4.1. Statistical Analysis of Objective Results

Table 2 shows a summary of the objective variables to be considered in the statistical analysis procedure. The first variable is related to the vehicle operation mode and will consider the time spent in automated mode, manual mode, and transitioning mode. The other two variables are related to safety. On one hand, the road departure events are extracted from the analysis of the lateral error of the vehicle with respect to the center of the right lane. On the other hand, the collision events are analyzed by the TTC measurement between the ego-vehicle and the oncoming vehicle in the left lane. After performing the official tests with 13 participants in the DiL simulator, the data were post-processed and analyzed in each category.

#### 4.1.1. Driving Mode

This KPI considers three driving modes: (1) manual (λ=0 Nm), (2) automated (λ ≥3.1 Nm), which includes overriding actions, and (3) transition (0 ≤λ≤3.1 Nm) when going from manual to automated mode or vice versa. For the analysis, the time in each driving mode is computed as the percentage of total time spent in each mode. Figure 10 summarizes the results for all driving sessions.

The time in automated mode reflects that in both L2 and SC the ADS was active for most of the driving session, with SC operating for around 10% more time in this state. This is attributed to the correction assistance that is not available in L2, as well as faster activation during lane returns (due to automatic transition instead of manual activation by pressing the button). Moreover, SC operates in manual mode half of the time compared to L2, which is a positive indicator, as the preferred state is to drive with some automation assistance and use manual mode only when necessary (overtaking the truck). Concerning the transition between driving modes, the percentage of time switching to manual is almost four times longer on SC than on L2, with an average duration of transitions of 0.67 s vs. 0.3 s, respectively. Longer transitions are smoother and avoid sudden lane departures. The percentage of time to transition to automated is almost seven times longer on SC than on L2, with an average duration of transitions of 0.72 s vs. 0.3 s, respectively. Longer transitions are smoother and reduce the number of lateral error peaks when returning to the lane.

#### 4.1.2. Road Departure Events

Unsafe events can be determined based on the vehicle position by analyzing the lateral error ($e_{y})$. It is done through a boxplot analysis and an event analysis. The boxplot analysis is shown in Figure 11. It is divided into two types of maneuvers. Maneuver A, represented by successful overtaking, and maneuver B, represented by aborted overtaking.

In maneuver A, it is of interest to study the maneuver peaks when both departing (A1) and returning to the lane (A2). In the case of A1, peaks when departing the lane show similar performance, though SC shows a smaller dispersion and whiskers (i.e., more consistent vehicle positioning in the left lane before overtaking the truck). In A2, SC improves the consistency when returning to the lane, and all the data are within the safety zone, showing no road departures. This proves the advantage of the automatic activation of the ALC system in comparison to the manual activation in the L2 baseline.

In addition, in maneuver B, it is of interest to study the maneuver peaks before the correction (B1), and after the correction (B2). In B1, SC shows that more than 50% of the time the vehicle was with most of the chassis within the lane, whereas in L2 more than 50% was off the lane; this difference is attributed to the system correction which is faster than the reactions of the human driver, in addition to a longer time to deactivate–as it only transitions to manual when it is safe and there is enough field of view. In B2, the SC dispersion is the smallest, with the vehicle wheels being kept within the lane (except for some outliers), while in L2, part of the 4th quartile trespasses the safety threshold.

In addition, an event analysis of the road-departure events is performed considering the four situations (A1, A2, B1, B2). Results are shown in Figure 12 (maneuver A) and Figure 13 (maneuver B).

In successful overtaking (A), SC reduces the departure off-lanes (A1) by half in comparison to L2 (mainly due to the smoother transitions from automated to manual). On the other hand, results show that when the driver returns to the lane after overtaking the truck, SC presents no return off-lanes (A2), while six departures are found with the L2 system (4% of total overtaking maneuvers). This result can be attributed to the advantage provided by the automatic transition to automated mode on SC rather than manual activation by pressing the button in L2.

Regarding aborted overtaking maneuvers (B), results show that the median position when using SC is below the departure threshold (2 m), while with L2 the median is above. In terms of departure off-lanes (B1), almost 70% of L2 events end with a lane departure, while SC reduces the off-lanes events to 48%. This improvement is due to the steering corrections that SC provides when detecting an oncoming vehicle. On the other hand, one might expect that this strong correction could result in an off-road event in the right lane. However, the return off-lanes (B2) analysis shows a similar number between L2 and SC (8% vs. 7%), suggesting that SC corrections do not increase the risk of an off-road event.

#### 4.1.3. Collision Events

To evaluate the collision-related events, the analysis procedure uses time-to-collision (TTC) between the vehicle in the right lane and the vehicle in the left lane. The calculation of TTC takes into account that the speed of both the ego-vehicle ($v_{x})$ and the speed of the vehicle in the left lane remains unchanged and is not infinite when part of the vehicle being driven is in the left lane. In Equation (7), the distance between the two closest points of the vehicles is defined by D.
(7)$$TTC=\frac{D}{v_{x}+v_{xl}};\;\;\;\;\;\;\;\;if\;e_{y}>1.5\;m$$

Additionally, the value of TTC represents different collisions event, as it is shown in Table 3.

**Crashes:** When the two vehicles collide after an overtaking attempt. When TTC = 0.**Near-misses:** Defined as “incidents in which no property was damaged, and no personal injury was sustained, but where, given a slight shift in time or position, damage or injury easily could have occurred”. Although there is no definitive threshold for considering near misses, the overall value is between 0.5 s and 1 s [30]. Since the scenario already considers a critical situation with the appearance of unexpected vehicles, the threshold of 0.5 s was preferred to capture a better difference between the systems.**Misses:** Those presenting a high probability of collision (excluding accidents and near-misses). The safety threshold for this KPI is 1.5 s, which is half of the minimum TTC that naturally occurs when the left vehicle appears at a very close distance (125 m); this is also a value used as a safety threshold in the literature [31,32].**Safe:** Those outside the first three indicators with respect to the total number of overtaking events.

Figure 14 shows the Time-Exposed-to-TTC (TET) as a percentage of the time. This is a global safety indicator for the systems. The percentage takes into account only the time in which both vehicles would cross if they would keep their current course (i.e., the total time for the calculation is not the time of the driving session but the time in which TTC ≠∞) [33]. In this sense, TET is defined as in Equation (8).
(8)$$\mathrm{TET}=\frac{\mathrm{time}\;(\mathrm{TTC}<\mathrm{Threshold})}{\mathrm{time}\;\left(\mathrm{TTC}\neq \infty \right)}$$

Overall, SC shows safer performance compared to L2 based on quartile analysis (median, maximum, and minimum values). For the L2 system, a TTC of 0.5 s is present in most participants driving sessions. However, SC shows the most remarkable result by ensuring a TTC > 1.5 s for all participants and events, except for two outliers (one accident and one near miss that occurred after overtaking the truck, as reported in Figure 15). This means that the SC mode improves road safety thanks to the protective function that applies steering corrections when an oncoming vehicle appears, according to the design of the shared controller and the arbitration system of Section 3.1.

In addition, an analysis of the collision events is performed categorizing different situations (i.e., accidents, near misses, misses, and safe events). Results are shown in Figure 15, which considers the total amount of the events, considering both departure and returning maneuvers.

Thanks to the correction capability of the SC mode, it allows keeping the vehicle under a safe condition for about 97% of the overtaking attempts–20% more than L2.

In terms of misses’ events, the results show that SC has a better safety indicator than L2 as it considerably reduces the number of events with TTC below 1.5 s (from almost 20% to around 2%).

For the near misses, there were seven events of this type for the L2 mode and only one for SC. The most valuable result is that no near miss behind the truck was reported for SC (compared to six for L2), which means that the correction functionality not only reduced but eliminated the near misses in that condition.

Finally, two accidents were reported after the experiment, one in L2 driving mode and one in SC. Both occurred after overtaking the truck and returning to the lane (i.e., the driver did not keep the required speed to finish the overtaking). One lesson from these results is that to further improve safety, the shared control system could be complemented with an intelligent longitudinal control during overtaking. In addition, one participant drifted driving under L2 mode, departing from the road and ending the driving session.

### 4.2. Statistical Analysis of Subjective Results

Subjective evaluations are collected by questionnaires administered to participants at the end of each driving session. The System Usability Scale [27,28] is used to explore more in-depth use of the system and satisfaction of the system. It consists of a 10-item questionnaire with five response options for respondents; from “Strongly agree” to “Strongly disagree”. The second scale administrated at the end of each scenario is the System Acceptance Scale of Van der Laan [29]. This scale investigates the acceptance of new technologies, and it consists of nine items, rated on a five-points scale, and includes two factors: usefulness of the system and satisfaction of the system.

#### 4.2.1. SUS

For the SUS–System Usability Scale–we asked participants to evaluate the system regarding the following aspects shown in Table 4.

Generally, participants reported they would like to use the SC system frequently and reported that the system was easy to use. The SC system obtained a higher score (SUS = 80 = A−) than the L2 systems (SUS = 76.3 = B), but both conditions obtain quite high values and both systems obtained results above the normative average (68). Individual questions results are shown in Figure 16. A first look shows that SC has a better usability score than L2 for 8 of the 10 questions. At SUS-08 and SUS-10, the L2 system is perceived as less cumbersome to use and less learning-intensive, which is to be expected since the SC mode integrates more features. Nevertheless, the ratings for SC are positive.

The confidence interval around the difference between the means of the two systems (3.7) does not rule out 0, and the *p*-value (*p* = 0.33) is not below the 95% confidence level (*p* < 0.05), so the results cannot say that there is a significant difference between the two systems. A look at the confidence intervals of the two systems shows that they have common ranges. With these means (80 and 76.3), increasing the number of samples to 51 would result in a *p*-value of 0.0488, which means that a higher number of samples could result in a significant difference between the two systems.

#### 4.2.2. SAS

After driving in the different conditions, participants also evaluated how the driving was experienced, using the System Acceptance Scale, by referring to the antonyms adjectives shown in Table 5.

Generally, participants evaluated all the different driving sessions positively. All values are included within the range from 3 (neutral value) to more than 4 (very positive) for both conditions. More in-depth, the SC system is evaluated similarly to the L2 system, and it is considered as more assisting, attentive, and desirable, while less annoying.

For a more concise evaluation of the results, the SAS methodology groups the odd fields with usefulness indicators, and even fields represent user satisfaction after testing the system. The scores range from −2 to 2, with 2 being the better result. The 2D graph shows the overall results of user acceptance, as shown in Figure 17. The SC system has better acceptance in both usefulness and satisfaction, with an overall score of 1.12 (compared to 0.73 for the L2 system).

## 5. Discussion

The interaction process with the driver, where human experiences are relevant, can be solved with Artificial Intelligence techniques, i.e., CNN and Deep Learning (DL), Genetic Algorithms (GA), and other “traditional” machine learning (ML) approaches. As mentioned by Na Du and colleagues [34], in conditional AD, drivers have difficulty taking over control when requested, especially if they are engaged in non-driving-related tasks (NDRTs) and they are requested to intervene from AD in different situations. Therefore, this topic is very important and broadly addressed in the literature. This section provides a discussion based on similar works–at least the ones the authors think are most relevant and newest, to the best of our knowledge.

### 5.1. Comparison with Similar Works

As mentioned in Section 1.4, to the knowledge of the authors, the present work is the first one studying a shared control interaction in the scenario of overtaking with oncoming traffic. The most similar work is presented by Walch et.al. [14]. It targets an analogous scenario, but with a different human–machine interaction, where human and automation cooperate only at the tactical level. To approve or cancel the overtaking maneuver, the driver uses a tactile panel, but the control actions are performed by the automation alone. In terms of the evaluation study, the metrics are focused on the quality of the different interaction strategies. On the contrary, our paper investigates cooperation mostly at the control level (both driver and automation guide the vehicle interacting through the steering wheel). In addition, the evaluation is mainly focused on the safety impact. Based on these differences and considering that these works do not share similar key performance indicators, a numerical comparison is not possible. The same is true for other related works that implemented shared control in overtaking maneuvers but did not consider oncoming traffic [18], or focused on the blind spot scenario instead [16]. Therefore, the comparison with similar works is presented in a qualitative manner rather than a quantitative one.

Yan et al. [35] investigate how to balance Lane Keeping (LK) performance and driving freedom when driver error occurs. Their approach is based on a safety evaluation strategy that assesses driver error and lane departure risk caused by driver error. Then, they proposed three models (for driver errors, cooperation, and shared authority), as well as a model predictive control (MPC)-based controller, implemented in MATLAB/SIMULINK, where they run numerical simulation tests. This work has some analogies with ours (e.g., cooperation approach, use of MPC, use of MATLAB-based simulations), but it is more focused on LK maneuvers in simpler scenarios.

Li and colleagues [36] propose a shared control driver assistance system based on the driving intention identification and situation assessment to avoid obstacles. They used an MPC (specifically, a constrained linear-time varying model predictive controller or LTV-MPC in short), a human driver’s driving intention recognition of the desired maneuvers, and a shared control fuzzy controller, which denotes the cooperative coefficient for the control authority between the controller and a human driver in different conditions. The investigated scenarios are related to obstacle avoidance and the cooperation is focused on the vehicle stability performance during these maneuvers. Although our experimental study includes more participants and the maneuvers are more complex (not only static but also dynamic obstacles, with overtaking), this work has nevertheless inspired us to improve our research, including the driver’s intention recognition (see the section related to the “next activities”).

Even the work of Benloucif et al. [37] deals with the concept of the shared control framework, established by a novel cooperative trajectory planning algorithm and a fuzzy steering controller. Unlike our work, this paper only addresses the driver–automation shared driving control for lane keeping and obstacle avoidance in a highway context, and no more complex maneuvers are considered. Moreover, the work focuses more on cooperative trajectory planning formulated on polynomial-based functions, while our work manages the conflict and cooperation at the control level using the MPC approach, which seems more robust and capable of best performance, in terms of prediction and accuracy [12].

All in all, the cooperative control authority allocation is addressed by many other works, being a crucial point for achieving the balance between the intervention of the human agent and the artificial agent. Thus, Zhang and colleagues [38] proposed a shared control scheme for LK with a fixed authority, while Guo et al. [39] designed an MPC-based shared steering control method. Both these methods (but also others can be considered, such as the works of Merah et al. [40] and Xing et al. [41]) ensure seamless control transfer between the system and the driver, but the authority for shared lateral control should be dynamic and well-shaped concerning drivers’ activity and driving task. The design principle shall take into account that the control authority should be given to the driver when being demanded, but it should be reduced on time if driver error caused the driving risk.

Among the two basic categories of control frameworks (coupled shared control [10] and uncoupled shared control [41]), we regarded the first one as relevant for haptics feedback control and human–machine interaction through force feedback. In particular, the coupled and cooperative control scheme achieves a proper balance between driving performance and driving freedom, and the dynamic authority allocation method adapts the arbitration to variations in driver error, lane departure, and velocity. From this perspective, researchers in [42,43] proved that haptic shared control can improve task and safety performance. However, when and how to intervene is still an open issue in the design of shared control systems (see also the work of Mars et al. [44]).

To sum up, our most relevant findings are related to safety improvement (which is also peculiar to this work) compared with other L2 commercial systems. In particular, we considered crucial aspects such as dynamic corrections, automatic activation to maintain safety conditions, specific HMI support, and so on. In addition, our tests involved “real” users and have been performed in dynamic scenarios, considering two-way roads for the evaluation of the shared control, which no other work to the knowledge of the author has addressed before.

### 5.2. Limitations and Future Works

The results achieved have some limitations: first, this study used time series data as model inputs, but without considering the complexity of sequence dependence among the data. In addition, it is not always easy to have a predetermined model structure based on the characteristics of variables, thus in some situations discerning membership function parameters by looking at data can be difficult (or even impossible). In these cases, rather than choosing the parameters associated with a given membership function arbitrarily, it is possible to tailor the membership function parameters to the input/output data. Moreover, to take the information contained in larger datasets on driving behaviors into account, we can use ensemble methods to improve the decision-making algorithms. Among various combinations of methodologies in soft computing, the one that can combine fuzzy logic and neurocomputing, leads to neuro-fuzzy systems. Within fuzzy logic, such systems play an important role in the induction of rules from observations: an effective method developed by Dr. Roger Jang for this purpose is called ANFIS (Adaptive Neuro-Fuzzy Inference System [45]), in which the learning algorithm tunes the membership functions of a Sugeno-type fuzzy inference system using the training input/output data. We will investigate this technique in our next step of research.

Another aspect is related to the experiments we have carried out, in which a limited number of participants has been recruited (for a matter of time and COVID-19 restriction protocols). Thus, in future work, this research will be extended in this sense.

Third, to make the decision more effective and accurate, some human factors should be taken into consideration–mainly the cognitive status of the driver, by including the driver monitoring system. More experiments can be conducted, with more people and analyze the different parameters that would affect the state of the driver such as food, time of driving, if they have slept well, etc.

Furthermore, again dealing with human-like behavior, the reference path for the shared control system will be updated by considering the prediction of the driver’s maneuver intention or the driver’s characteristics in the path planning module, such as the driving style. Sometimes the human driver tends to make wrong decisions due to poor driving states such as distracted and fatigued driving (hence, these two aspects should be considered together). To contribute to the prevention of the potential traffic accident, a future step of our research will be to investigate a specific framework for the arbitration and sharing control, including the driving intention and situation assessment. Finally, another “next step” will be the implementation of the arbitration and sharing control system in a real demonstrator vehicle, thus integrating the developments made in the DiL simulator platform into an industrial-relevant driving environment.

## 6. Conclusions

This section provides some conclusive remarks, focusing particularly on a general summary and, above all, on the lesson learnt after this project.

### 6.1. Summary

The present paper adopts the shared control scheme as the fundamental design framework of a cooperative agent for supporting the driver. We explored the advantages of the shared control adaptive co-pilot over commercially available autopilots (i.e, SAE L2 systems combining ALC with ACC functions), with a particular focus on situations where the drivers are encouraged to support automation. The demonstration of the system’s abilities is given in a specific use-case and test scenario, in which the AD system and the human driver act as a team, supporting each other to cover and overtake their single drawbacks and weak points. Considering the selected KPIs, our proposed solution shows better performance of the combined team (human–automation), concerning the baselines we considered (ADAS working separately, namely SAE-L2): the critical situations are reduced and there is also an improvement in the driving performance.

Thereby, given the achieved results, we can say that the answer to the question of the title (“Can shared control improve overtaking performance?”) is positive because the proposed framework can enhance the safety and user acceptance of the proposed system for the overtaking maneuver and thus of the driving task.

### 6.2. Lessons Learnt

Prospective developments concerning what we have learnt from this experience are discussed in this section.

First, the effectiveness of the proposed arbitration and shared control algorithms are demonstrated with both objective and subjective evaluations, but it is necessary to reduce driver–automation conflict as much as possible. In this context, a proper HMI is essential to help drivers to better understand the automation’s actions (and thus to reduce further conflicts). In this sense, it has been also crucial to also involve the “end-user” in the data collection and development phases (especially in those scenarios where the adaptation between human and machine agents is fundamental for safe and effective operations) to investigate their initial acceptance and to guarantee a good diffusion and penetration of these systems on the market later on.

Second, the proposed algorithms represent a successful technology to enlarge ODD and enhance the robustness of standalone automated driving systems, because adapting the level of driver involvement to the driving context can enable a smoother traded control mechanism, leading to a higher user acceptance of SAE L2/L3+ systems. In fact, the obtained results in our use case show a positive impact on the shared control mode for driving safety, performance, and comfort, which has been corroborated by the user acceptance tests. The proposed solution can encourage the idea of being a “team” and provide benefits to the driving task that would help to avoid “automation irony” (as the driver has a sense of being responsible for driving even under automation support). At the same time, drivers benefited from the automation capabilities to intervene during unsafe events, improving driving safety while reducing the driver-required efforts.

Therefore, to ultimately ensure valid and reliable application of our framework in practice, it should continue to be developed and improved upon in future works.

## Figures and Tables

**Figure 1 sensors-22-09093-f001:**
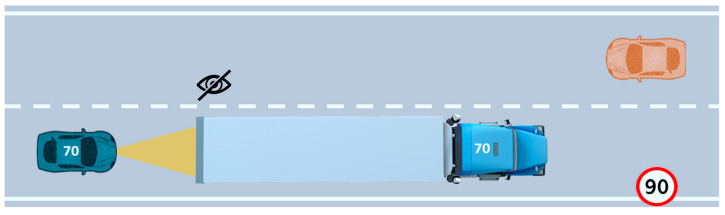
The overtaking scenario on a road with oncoming traffic.

**Figure 2 sensors-22-09093-f002:**
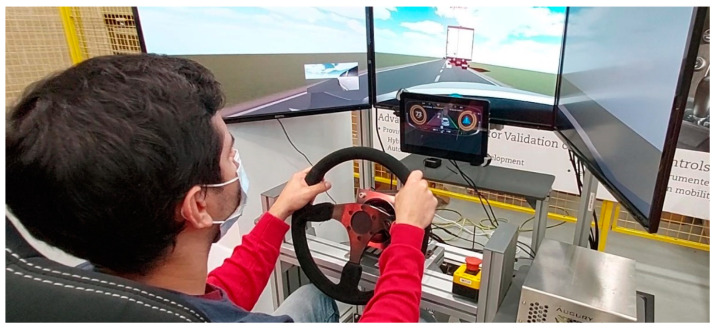
DiL simulator for AD applications with Dynacar vehicle simulation software in screens, and Re:Lab HMI as the instrument cluster.

**Figure 3 sensors-22-09093-f003:**
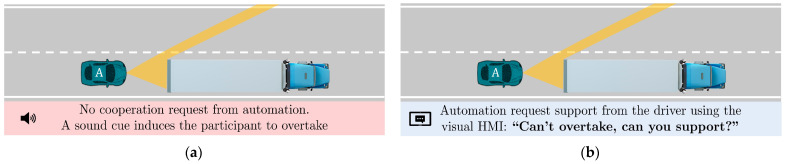
Baseline L2 vs. SC system. A comparison of functionalities in four aspects: (**a**,**b**) the visual HMI interaction, (**c**,**d**) the correction maneuver, (**e**,**f**) the transition from automated to manual, and (**g**,**h**) the transition from manual to automated.

**Figure 4 sensors-22-09093-f004:**
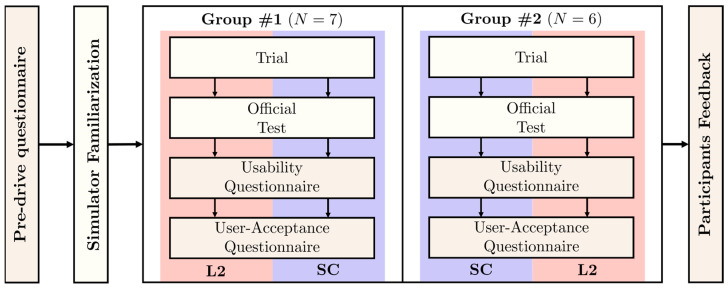
Outline of the within-subject driver study.

**Figure 5 sensors-22-09093-f005:**
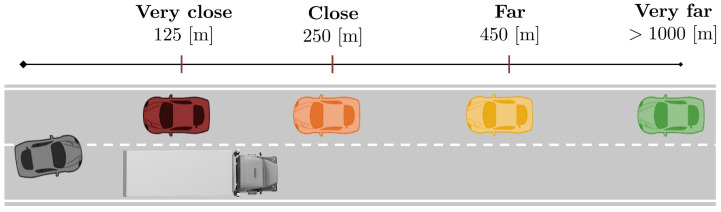
Upcoming vehicle events at four pre-defined distances.

**Figure 6 sensors-22-09093-f006:**
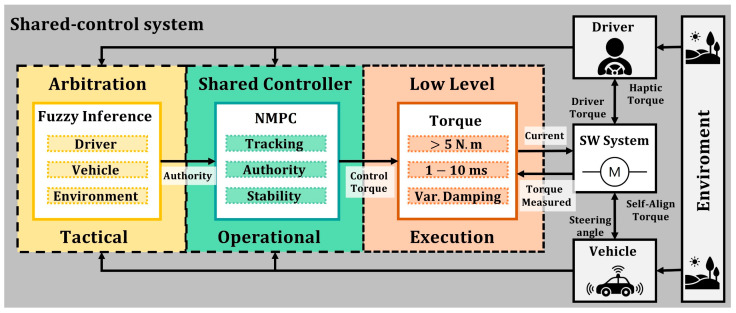
Shared control framework for Automated Vehicles.

**Figure 7 sensors-22-09093-f007:**
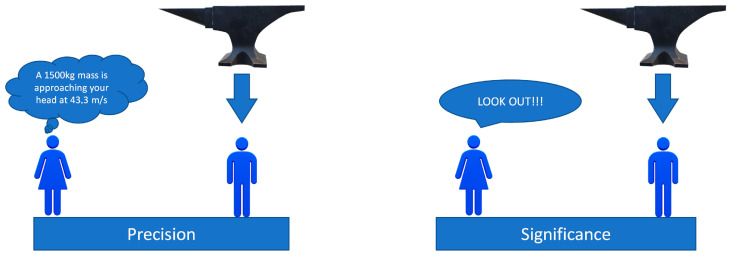
Meaning of FIS, concepts of precision, and significance in real world.

**Figure 8 sensors-22-09093-f008:**
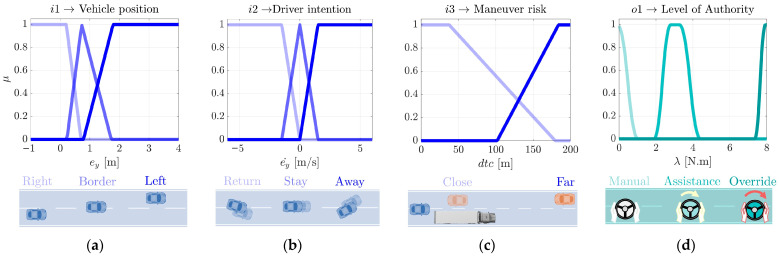
Fuzzy logic membership functions of arbitration module: (**a**) input 1 is the lateral error with respect to the right lane, (**b**) input 2 is the derivative of the lateral error and combined with input 1 helps to detect the driver lane change intention, (**c**) input 3 represents the risk of the maneuver, (**d**) the output of the arbitration system, which decides based on the three inputs, if the steering assistance is null (Manual), enough to help (Assistance) or strong enough to avoid the driver maneuver (Override).

**Figure 9 sensors-22-09093-f009:**
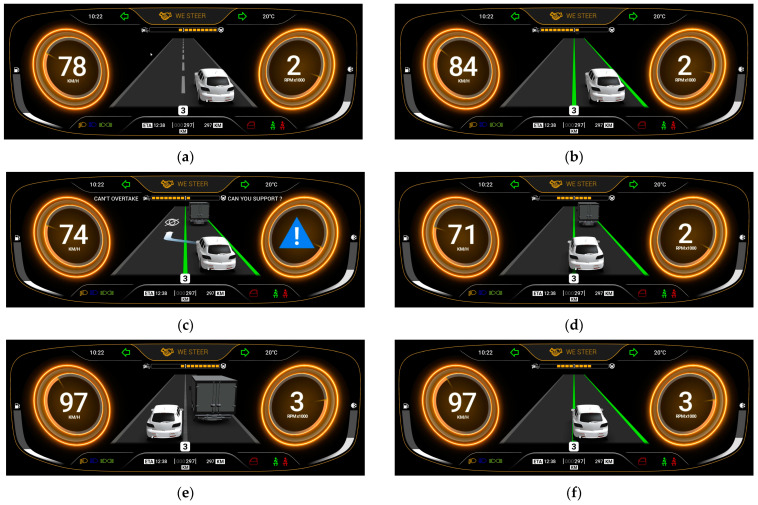
Sequence of overtaking using SC from the HMI view: (**a**) vehicle in manual mode, (**b**) vehicle in automated mode at maximum speed, (**c**) cooperation suggested by automation, (**d**) driver initiates lane change and transition to manual starts, (**e**) transition to manual completed to perform overtaking, (**f**) driver returns and automated mode is activated automatically.

**Figure 10 sensors-22-09093-f010:**
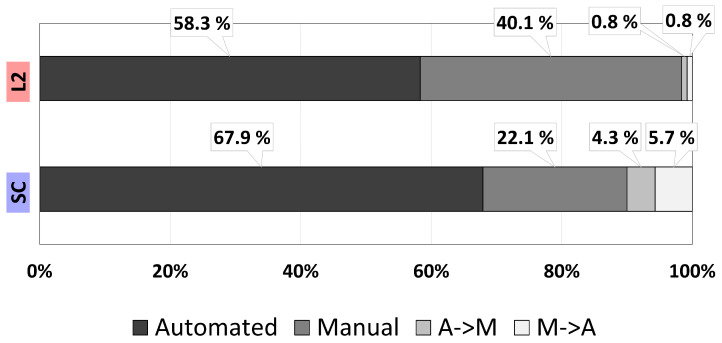
Percentage of time in each driving mode during the driving sessions.

**Figure 11 sensors-22-09093-f011:**
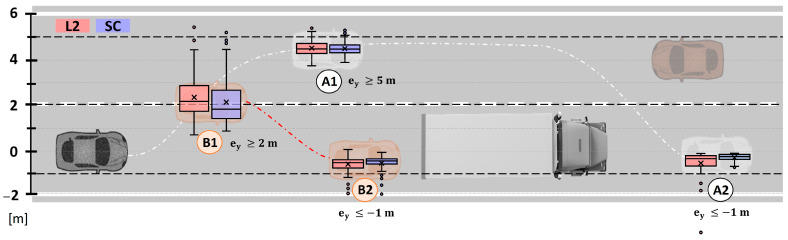
Boxplot analysis of the road-departure events. Two events are detailed: A-events related to those overtaking maneuvers finished, and B-events which are of those overtaking maneuvers aborted.

**Figure 12 sensors-22-09093-f012:**
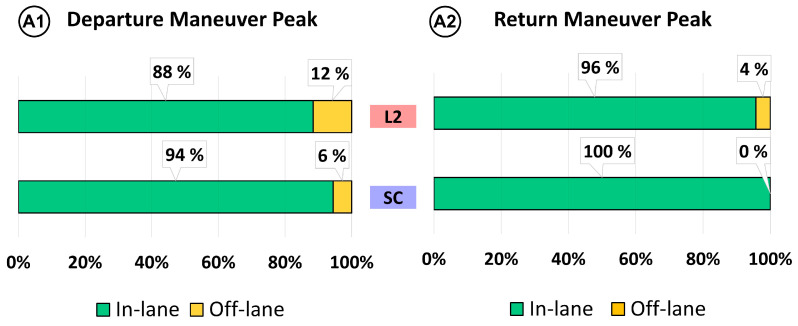
Event analysis of successful overtaking (maneuver A).

**Figure 13 sensors-22-09093-f013:**
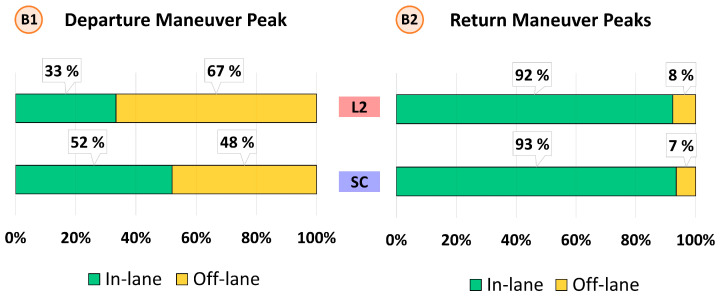
Event analysis of aborted overtaking (maneuver B).

**Figure 14 sensors-22-09093-f014:**
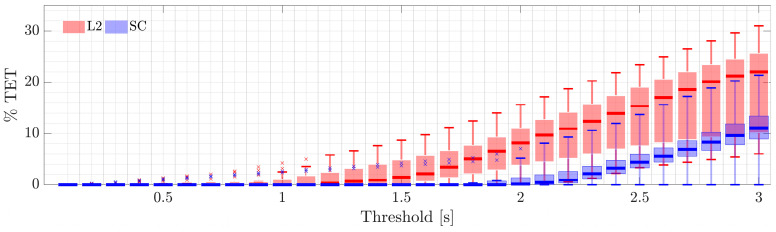
TET considering different TTC thresholds (between 0 and 3 s).

**Figure 15 sensors-22-09093-f015:**
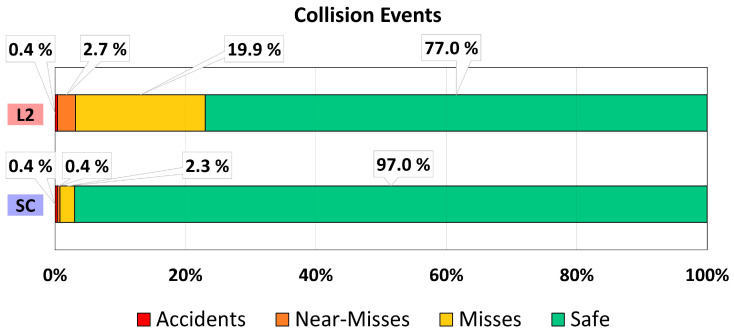
Analysis of collision-related events (TTC).

**Figure 16 sensors-22-09093-f016:**
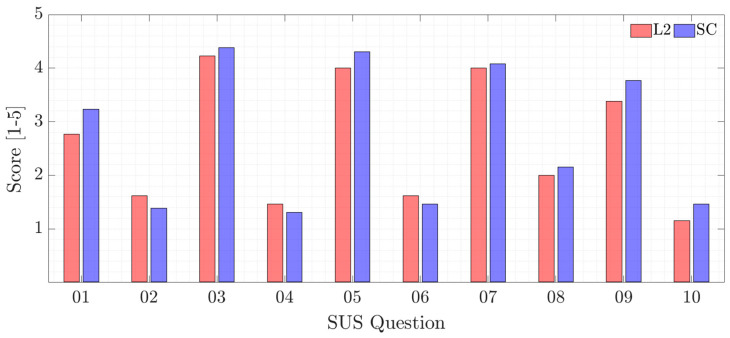
System Usability Scale individual questions results.

**Figure 17 sensors-22-09093-f017:**
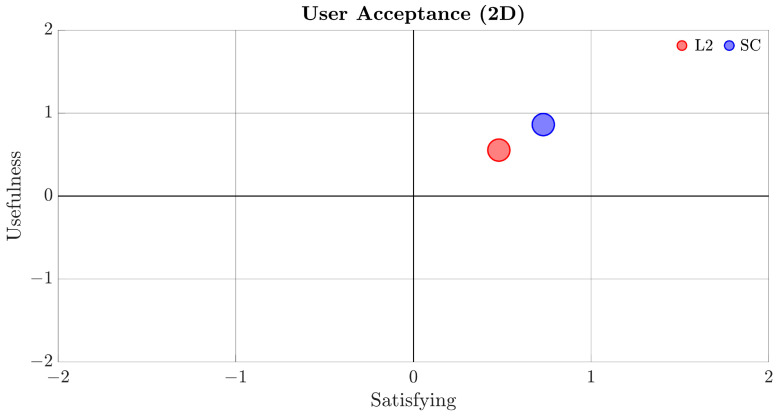
System Acceptance Scale results (Satisfying vs. Usefulness).

**Table 1 sensors-22-09093-t001:** The FL rules of the arbitration system for SC mode, for the cooperative overtaking maneuver. The rules include three inputs: the lateral error ($i_{1}$), the lateral error derivative ($i_{2}$), and the distance to collision ($i_{3}$), and also the output ($o_{1}$) which is the level of steering assistance or torque intensity, which is called the haptic authority (λ).

$i_{1}=e_{y}\;\to$	Right			Border			Left		
$i_{2}=\dot{e_{y}}\;\to$	Return	Stay	Away	Return	Stay	Away	Return	Stay	Away
**Left**	Assist	Assist	Assist	Assist	Assist	Override	Override	Override	Override
**Right**	Assist	Assist	Assist	Assist	Assist	Manual	Manual	Manual	Manual
$i_{3}=dtc\;↑$	$o_{1}=\lambda \;↑$								

**Table 2 sensors-22-09093-t002:** Variables for statistical analysis of objective results.

Analyzed Variable	Category	Main Variable
Driving mode	Operation mode	Time in each mode
Road departure events	Safety	Lateral error $\left(e_{y}\right)$
Collision events	Safety	Time-to-collision (TTC)

**Table 3 sensors-22-09093-t003:** Collision events related to the TTC value.

Event	TTC to the Oncoming Vehicle
Crashes	TTC = 0 s
Near misses	0 s < TTC < 0.5 s
Unsafe	0.5 s ≤ TTC ≤ 1.5 s
Safe	TTC ≥ 1.5 s

**Table 4 sensors-22-09093-t004:** System Usability Scale Questionnaire.

#	Question
01	I think that I would like to use this system frequently.
02	I found the system unnecessarily complex.
03	I thought the system was easy to use.
04	I think I would need the support of a technical person to be able to use this system.
05	I found the various functions in this system were well integrated.
06	I thought there was too much inconsistency in this system.
07	I would imagine that most people would learn to use this system very quickly.
08	I found the system very cumbersome to use.
09	I felt very confident using the system.
10	I needed to learn a lot of things before I could get going with this system.

**Table 5 sensors-22-09093-t005:** System Acceptance Scale Questionnaire.

Useful	o	o	o	o	o	Useless
Pleasant	o	o	o	o	o	Unpleasant
Bad	o	o	o	o	o	Good
Nice	o	o	o	o	o	Annoying
Effective	o	o	o	o	o	Superfluous
Irritating	o	o	o	o	o	Likeable
Assisting	o	o	o	o	o	Worthless
Undesirable	o	o	o	o	o	Desirable
Raising alertness	o	o	o	o	o	Sleep-inducing

## Data Availability

Upon request to the corresponding author.

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
