# Peer review of "Can Shared Control Improve Overtaking Performance? Combining Human and Automation Strengths for a Safer Maneuver"

_sensors, 2022, doi:10.3390/s22239093_

Round 1
Reviewer 1 Report
The authors propose a shared control scheme to improve the overtaking performance in automatic driving. The research is meaningful, however, the author's description of the method is still unclear. In particular, compared with similar studies, the advantages of this paper need to be clear.
And the following issues need to be revised.
1. ‘5.1 Comparison with similar works’:When compared with other studies, whether there are quantitative indicators to evaluate the superiority of the proposed model?
2. When there are multiple sub figures, the sub name of the figure should be marked.
3.Normalize the References.
4.The author's description of the method is unclear and needs to be further clarified.
5.At present, there have been studies on sharing mechanisms such as drivers, cars and roads. What are the differences and advantages between manuscripts and such studies?
Author Response
Dear Editor and Reviewers,
We would like to thank you for the time and effort spent through all the revision process, and especially for all the meaningful inputs and helpful feedback. The article was carefully revised following all the reviewer’s comments.
Best regards,
The authors
REVIEWER #1 COMMENTS
The authors propose a shared control scheme to improve the overtaking performance in automatic driving. The research is meaningful, however, the author's description of the method is still unclear. In particular, compared with similar studies, the advantages of this paper need to be clear.
And the following issues need to be revised.
1) 5.1 Comparison with similar works’: When compared with other studies, whether there are quantitative indicators to evaluate the superiority of the proposed model?
We highly appreciate the reviewer’s comment. Authors consider that this is a valid concern as comparison with similar works are commonly presented with numerical comparisons of the results. In this case, as we mentioned in Section 1.4, to the knowledge of the authors, our work is the first one that proposes a shared control strategy in the scenario of overtaking in roads with oncoming traffic. There is a similar work which we tried to compare with, but the focus of the study was more on the interaction than in the safety impact, and we could not find any common KPI to perform a numerical comparison, neither with other related works. We have included the following paragraph in Section 5.1.
As mentioned in Section 1.4, to the knowledge of the authors, the present work is the first one studying a shared control interaction in the scenario of overtaking with oncoming traffic. The most similar work is presented by Walch et.al. [13]. It targets an analogous scenario, but with a different human-machine interaction, where human and automation cooperate only at the tactical level. To approve or cancel the overtaking maneuver, the driver uses a tactile panel, but the control actions are performed by the automation alone. In terms of the evaluation study, the metrics are focused on the quality of the different interaction strategies. On the contrary, our paper investigates the cooperation mostly at the control level (both driver and automation guide the vehicle interacting through the steering wheel). In addition, the evaluation is mainly focused on the safety impact. Based on these differences and considering that these works do not share similar key performance indicators, a numerical comparison is not possible. The same is true for other related works that implemented shared control in overtaking maneuvers but did not consider oncoming traffic [17] or focused on the blind spot scenario instead [15]. Therefore, the comparison with similar works is presented in a qualitative manner rather than a quantitative one.
2) When there are multiple sub figures, the sub name of the figure should be marked.
We thank the reviewer for this comment that help us to improve the quality of the article presentation. Figures with sub-figures have been marked now according to the journal format. Figure 3, 8, and 9.
3) Normalize the References.
We appreciate the level of detail of the reviewer when revising the article. It is true some references were not according to the journal format, and it has been now checked and fixed according to the journal guidelines (https://www.mdpi.com/authors/references)
4) The author's description of the method is unclear and needs to be further clarified.
Authors agree with the reviewer in that the methodology of the experiment missed important details. We have now added a new figure that graphically explain the procedure used for the driver study. The following text and Figure 4 were added in Section 2.3.3.
Participants filled out a pre-drive questionnaire related to demographic data and driving profiles. Also, they familiarized themselves with the simulator, taking a driving session in manual mode. The study was a within-subject experiment, and the order of execution of both systems was different according to two corresponding participant groups. Figure 4 (see in manuscript) shows the schematic of the experiment procedure including the order of application of questionnaires.
5) At present, there have been studies on sharing mechanisms such as drivers, cars and roads. What are the differences and advantages between manuscripts and such studies?
Authors thank the reviewer for this comment. In our paper we use the term “shared control” as a very particular mode of human-machine interaction which is that they “work in the same task and at the same time”, particularly at the decision and control level as defined in references [9,10]. In this sense, this concept of sharing the execution of a task is not related to similar terms implying sharing of resources, such a cars and roads, which is what we understand from reviewer’s comment.
An update version of the manuscript is sent as an attached file

Reviewer 2 Report
This paper manuscript explores the advantages of the shared control adaptive co-pilot over commercially available SAE L2 systems, and an improvement in the driving performance is shown by achieved results. It’s valuable to the readers in automated driving domain.
There are still some minor points can be improved, for example:
The traffic safety reports from National Highway Traffic Safety Administration (NHTSA) mentioned in 1.1 The road safety problem shall be cited.
However, if the authors can take more reasonable safety metrics, e.g., PTTC (potential time to collision), TET (time exposed TTC) into account, and compare them with current metric (TCC), it will be perfect.
Quality of some figures shall be improved, e.g., Figure 9-12, 14, 16.
Author Response
Dear Editor and Reviewers,
We would like to thank you for the time and effort spent through all the revision process, and especially for all the meaningful inputs and helpful feedback. The article was carefully revised following all the reviewer’s comments.
Best regards,
The authors
REVIEWER #2 COMMENTS
This paper manuscript explores the advantages of the shared control adaptive co-pilot over commercially available SAE L2 systems, and an improvement in the driving performance is shown by achieved results. It’s valuable to the readers in automated driving domain.
There are still some minor points can be improved, for example:
1) The traffic safety reports from National Highway Traffic Safety Administration (NHTSA) mentioned in 1.1 The road safety problem shall be cited.
We thank the reviewer for noticing this missing reference. We have added the reference, but also updated the numbers according to a most recent report. This can be found in Section 1.1.
The traffic safety reports from National Highway Traffic Safety Administration (NHTSA) have registered more than 35,000 injury crashes, where around 8% correspond to crashes due to driver misbehavior. As a result, over 3,100 people died, and another 320,000 people were injured [1].
[1] National Center for Statistics and Analysis. Distracted Driving 2020; (Research Note. Report No. DOT HS 813 309). National Highway Traffic Safety Administration, 2022;
2) However, if the authors can take more reasonable safety metrics, e.g., PTTC (potential time to collision), TET (time exposed TTC) into account, and compare them with current metric (TCC), it will be perfect.
Authors appreciate the suggestion from the reviewer, as it is pertinent to our study and can improve the discussion of the results.
In relation to the TET, when getting the definition from reference [a], we noticed that this is equivalent to the PTTC (Proportion of TTC) that we defined in equation (9). The only difference is that we presented this value as percentage of time. Now, we think using the terminology proposed by the reviewer is more appropriate, thus, we changed the term PTTC by TET (%) in the corresponding text and Figures of Section 4.2.3.
[a]. Minderhoud, M. M., & Bovy, P. H. (2001). Extended time-to-collision measures for road traffic safety assessment. Accident Analysis & Prevention, 33(1), 89-97. https://doi.org/10.1016/S0001-4575(00)00019-1
Figure 13 shows the Time-Exposed-to TTC (TET) as a percentage of the time. This is a global safety indicator for the systems. The percentage takes into account only the time in which both vehicles would cross if they would keep their current course (i.e., the total time for the calculation is not the time of the driving session but the time in which TTC ≠∞) [33]. In this sense, TET is defined as in Eq. 9.
Concerning the PTTC, when revising the reference where PTTC is defined [b], we noticed that it is more related to the TTC with respect to the leading vehicle, but our metric is related to the TTC with respect to the oncoming vehicle. As in our study neither the oncoming vehicle nor the leading vehicle are changing behavior (driving at constant speed), we will keep the reviewer's suggestion for future works when we also consider the TTC with the leading vehicle and take into account a double risk of collision (lateral and frontal).
[b].https://www.researchgate.net/publication/242237773_Traffic_Conflict_Analysis_using_Vehicle_Tracking_System_with_Digital_VCR_and_New_Conflict_Indicator_under_High_Speed_and_Congested_Traffic_Environment
3) Quality of some figures shall be improved, e.g., Figure 9-12, 14, 16.
We are thankful for this comment as many figures were degraded in quality due to a misconfiguration of the Word document. Now, all figures have been checked and those with poor quality were improved.

Round 2
Reviewer 2 Report
After the revise, I believe the manuscript in current version has been sufficiently improved to warrant publication.